# Distinct p63 and p73 Protein Interactions Predict Specific Functions in mRNA Splicing and Polyploidy Control in Epithelia

**DOI:** 10.3390/cells10010025

**Published:** 2020-12-25

**Authors:** Julian M. Rozenberg, Olga S. Rogovaya, Gerry Melino, Nickolai A. Barlev, Alexander Kagansky

**Affiliations:** 1Laboratory of Molecular Oncology, Moscow Institute of Physics and Technology, 141701 Dolgoprudny, Russia; gm614@mrc-tox.cam.ac.uk (G.M.); nick.a.barlev@gmail.com (N.A.B.); 2Koltzov Institute of Developmental Biology, Russian Academy of Science, 119334 Moscow, Russia; Rogovaya26f@gmail.com; 3MRC Toxicology Unit, University of Rome Tor Vergata, 00133 Rome, Italy; 4Institute of Cytology, Russian Academy of Science, 194064 Saint-Petersburg, Russia; 5Orekhovich Institute of Biomedical Chemistry, 119435 Moscow, Russia; 6School of Biomedicine, Far Eastern Federal University, 690920 Vladivostok, Russia

**Keywords:** p53, p63, p73, protein interactions

## Abstract

Epithelial organs are the first barrier against microorganisms and genotoxic stress, in which the p53 family members p63 and p73 have both overlapping and distinct functions. Intriguingly, p73 displays a very specific localization to basal epithelial cells in human tissues, while p63 is expressed in both basal and differentiated cells. Here, we analyse systematically the literature describing p63 and p73 protein–protein interactions to reveal distinct functions underlying the aforementioned distribution. We have found that p73 and p63 cooperate in the genome stability surveillance in proliferating cells; p73 specific interactors contribute to the transcriptional repression, anaphase promoting complex and spindle assembly checkpoint, whereas p63 specific interactors play roles in the regulation of mRNA processing and splicing in both proliferating and differentiated cells. Our analysis reveals the diversification of the RNA and DNA specific functions within the p53 family.

## 1. Introduction

The p53-family is a group of proteins, which consists of p53, p63, and p73 [1,2]. p63 and p73 are paralogs of p53 and have similar structure: transactivation (TA) at the N-terminus following by DNA-binding, nuclear localization, tetramerization domains and sterile alpha motif (SAM) domain, a specific feature of the p63 and p73 at the C-terminus (Figure 1) [1,2,3].

p53, p63 and p73 proteins are represented by multiple isoforms that regulate each other’s transcription and functions [4,5,6,7,8,9]. Full length p73 and p63 contain TA domain and alternative intragenic transcriptional start sites between p63/p73 exons 3 and 4 produce N-truncated “DN” isoforms lacking the TA domain. Full length “-alfa” isoforms contain SAM domain and alternative C-terminal splicing produces beta, gamma, etc., isoforms lacking the SAM domain. Thus, an isoform nomenclature “TAp73α” means that a p73 isoform contains N-terminal TA domain with full length SAM domain at the C-terminal. Accordingly, DNp73β and above (gamma, zeta etc.) means that the activation domain is missing and the SAM domain is truncated. Thereby, combination of N- and C-terminal variants creates a variety of isoforms (Figure 1) [3,10].

In the full length TAp73α and TAp63a, N-terminal TA domain acts as the transcriptional activator whereas C- terminal containing SAM and inhibitory domains repress TA activity [11,12]. Generally, isoforms lacking TA domain DNp63 and DNp73 repress activity of isoforms containing TA domain [4,7]. Also, p73 isoforms can act as conditional activators or repressors [13,14]. For example, both TAp73β and DNp73β but neither p53, TAp73α, TAp63α, nor TAp63β induce the expression of caspase-2S [13]. Similarly, TAp73α co-activates c-Jun whereas TAp73β can act as a repressor of c-Jun mediated transactivation [14,15].

In particular, DNA-binding domain is the most conserved structural element across the p53-family (Figure 4), leading to identical DNA recognition [16,17,18,19], underlining the functional relatedness of these proteins as transcription factors. p53 functions as a transcription factor in charge of the activation of genes responsible for the cell-cycle arrest in G1 and G2/M checkpoints and apoptosis induction. At the same time, both p73 and p63 are known for regulation of the p53-responsive genes: MDM2, IGFBP3, p21 [20]. However, apart from the large amount of evidence that p53, p63 and p73 proteins have overlapping and interdependent functions in the apoptosis control and genome stability [21], p63 and p73 also have distinct specific roles in developmental processes [22,23,24,25,26], DNA repair [27,28,29] and ciliogenesis [30,31,32].

Specifically, p63 and p73 play distinct roles in epithelial morphogenesis. The p63 is a major player in skin biology and its role in epidermis was thoroughly investigated during the last two decades [33,34,35,36,37,38,39].

The p63 knockout in mice leads to E1 lethality and pups born with non-differentiated skin [33,34]. Total p63 knockout has been rescued by specific p63 isoforms expression under the control of KRT5 promoter [38]. Double TAp63a/DNp63a rescue showed regions of differentiated skin, while p63-/TAp63 did not rescue, whereas p63-/DNp63 showed only basal keratinocytes formation. Consistently, DNp63 regulated basal undifferentiated keratinocytes marker KRT14, whereas TAp63 regulated markers of differentiated keratinocytes (Ets-1, KRT1, transglutaminases, involucrin) [38]. In addition, p63 function is required for the establishment of the cell division perpendicular to the basal membrane at E18.5 day of the embryo [35,40]. Moreover, both β1-integrin and laminin are distributed around the cells and not localised to the basal membrane in the p63 knockout mice [40].

In contrast, the p73 knockout mice are characterised by runting phenotype, hippocampal dysgenesis and hydrocephalus, sterility, chronic inflammation and infection in the lungs, sinus, and ears [41]. Total p73 knockout mice do not have profound skin phenotype except the somewhat slower wound-healing response [42]. TAp73 knockout mice demonstrate sterility, hippocampal dysgenesis, hydrocephalus, premature aging, genomic instability, and increased frequency of tumors [24]. Finally DNp73 knockout causes neurodegeneration, specifically hippocampal dysgenesis and hydrocephalus [43].

In the human skin, some keratinocytes fail to divide and become polyploid upon differentiation that can be induced by the DNA damage and is associated with overexpression of Cyclin E [44,45]. Using fluorescence in situ hybridization for DNA, it was shown that human skin keratinocytes become multinucleated with their genomes increasing up to 4 fold [46,47].

Intriguingly, p73 displays a very specific localization to basal epithelial cells in all human tissue where it’s localization was determined by immunohistochemistry. In contrast, the p63 is expressed in both basal and differentiated cells [42,48].

The role of p63 in polyploidy has not been evaluated, however, the presence of p63 in differentiating keratinocytes suggests that p63 is not interfering with the polyploidy of these cells. At the same time, p73 suppresses polyploidy and aneuploidy in the absence of functional p53 [23,49]. Also, cells from TAp73−/− mice exhibit genomic instability associated with the enhanced aneuploidy [24]. Therefore, it is possible that the differential presence of the p63 and p73 in the differentiated and basal cells is related to the strict polyploidy control by the p73. Specifically, the absence of p73 allows cells to loosen the mitotic spindle assembly checkpoint and hence become polyploid and multinucleated.

Here, we systematically analyzed the literature describing protein interactions and functions of p63 and p73, trying to reveal their distinct functions that could be a physiological basis for their differential distribution in epithelial organs.

What we found is that p73 and p63 cooperate in the genome stability surveillance in proliferating cells with the specific p73 function in the APC/C and spindle assembly checkpoint. Unexpectedly, our survey suggests a distinct function of the p63 in the regulation of mRNA processing and splicing in both proliferating and differentiated cells.

## 2. Materials and Methods

Immunohistochemistry images at Figure 2 are from the Protein Atlas v19.3: http://www.proteinatlas.org [50].

Links to specific images are below:

Figure 2A: https://www.proteinatlas.org/ENSG00000078900-TP73/tissue/skin#img

Figure 2B: https://www.proteinatlas.org/ENSG00000073282-TP63/tissue/skin#img

Figure 2C: https://www.proteinatlas.org/ENSG00000148773-MKI67/tissue/skin#img

Figure 2D: https://www.proteinatlas.org/ENSG00000073282-TP63/tissue/nasopharynx#img

Figure 2E: https://www.proteinatlas.org/ENSG00000078900-TP73/tissue/nasopharynx#img

Figure 2F: https://www.proteinatlas.org/ENSG00000148773-MKI67/tissue/nasopharynx#img

Figure 2G: https://www.proteinatlas.org/ENSG00000073282-TP63/tissue/oral+mucosa#img

Figure 2H: https://www.proteinatlas.org/ENSG00000078900-TP73/tissue/oral+mucosa

Figure 2I: https://www.proteinatlas.org/ENSG00000148773-MKI67/tissue/oral+mucosa

Figure 2J: https://www.proteinatlas.org/ENSG00000073282-TP63/tissue/bronchus#img

Figure 2K: https://www.proteinatlas.org/ENSG00000078900-TP73/tissue/bronchus

Figure 2L: https://www.proteinatlas.org/ENSG00000148773-MKI67/tissue/bronchus

Data from the BioGrid database https://thebiogrid.org/, version 1.1.7 [51] was downloaded for human tp53, tp63 and tp73. Only physical interactions were included and common and distinct protein interactions were calculated using excel (Appendix A) and a proportional Euler diagram was used for visualization [52] http://www.eulerdiagrams.org/eulerAPE/.

Resulting lists were submitted to the Metascape 3.0 software and analyzed with default settings https://metascape.org/gp/index.html#/main/step1 [53]. In brief, Metascape extracts a protein interaction network from the list provided by the user and identifies densely-interacting complexes by iterations of the implementation of the molecular complex detection (MCODE) algorithm [54]. Further, for annotation of their biological roles, Metascape applies pathways and processes enrichment analysis to identified networks of protein interactions and annotated the top three enriched terms [53]. The Log_10_(*p*) values for top annotations of the identified putative protein complexes are summarized in the Appendix A.

Single cell RNA-seq data from the normal human epidermis was downloaded [55] and aligned with protein interaction dataset using gene names (Appendix A).

Post-translational p63 and p73 modifications were obtained from the dbPTM database http://dbptm.mbc.nctu.edu.tw/index.php [56].

Alignment of p73 (O15350) and p63 (Q9H3D4) protein sequences was performed by NCBI COBALT tool https://www.ncbi.nlm.nih.gov/tools/cobalt/cobalt.cgi?CMD=Web.

## 3. Results

### 3.1. p63 and p73 Distribution in the Normal Epithelial Tissues

According to the human protein and RNA expression atlas http://www.proteinatlas.org [50], the p53 protein could be detected at extremely low levels in the oral mucosa, oesophagus, urinary bladder, skin, and tonsils, and this expression is variable across samples and antibodies. In contrast, p63 and p73 proteins are consistently expressed. Moreover, proteins exhibit a characteristic expression pattern. In the skin, p63 levels are the highest in the basal cells, but also detected in the squamous, more differentiated cells whereas p73 is found only in the basal cells (Figure 2A,B). Similarly, in the nasopharynx (Figure 2D,E), vagina (not shown) and oral mucosa (Figure 2G,H) the p73 is restricted to the basal cells and p63 is expressed throughout the basal epithelium and differentiated cells. The p73 expression can be detected in a few cells of the basal layer of the salivary glands lumens, whereas p63 is determined in the different cells (not shown).

In the bronchial epithelium of the lungs, p63 and p73 staining patterns seem to be similar (Figure 2J,K). Whereas p63 antibody strongly stains these tissues, with the different staining patterns than described above, it also stains myoepithelial cells of the breast, urinary bladder, epididymis, seminal vesicles, prostate, and placenta. Interestingly, in these organs, p63 stained cells immediately next to the basal membranes, resembling the p73 staining pattern in the organs expressing both p63 and p73.

In vivo, mitosis mostly occurs in the basal layer, where the p73 is, but a few cells go through mitosis just above the basal layer (Figure 2C,F,I,L) marking the last asymmetric division, leading to the terminal differentiation for one cell and an ability to proliferate for another [35,55,57].

How do these data correlate with RNA expression levels of p63 and p73?

According to GTEx portal RNA-seq data, expression of p73 is quite tissue specific, with relatively high levels of DNp73 in epithelial tissue: skin, vagina, esophagus (similar to the protein Atlas data) and TAp73 is expressed in the brain https://www.gtexportal.org/home/gene/TP73.

Similarly to the immunohistochemistry images, the level of tp73 expression in skin is 7.8 transcripts per million (TPM), whereas the level of tp63 is 127 TPM and tp53 is 36 TPM (Appendix A). In comparison, MDM2 is 11 TPM and MDM4 is 24 TPM and CDKN1A (p21) is 225 TPM. According to the single cell RNA-seq of the human skin, the p73 RNA is under detection threshold [55]. However, to release single cells, the skin is digested at 37 °C for 2 h, which can lead to RNA degradation, especially for the unstable transcripts [58,59].

Given the low p73 RNA levels in skin and other epithelial organs, the question arises: how reliable is p73 antibody staining? We have thoroughly analyzed the available data using different antibodies and conclude that p73 expression has the same characteristic patterns in skin and other epithelial organs, as defined by experiments using antibodies raised against distinct p73 epitopes in independent studies.

Analysis of the literature and web resources revealed that there are 4 different antibodies against different epitopes that generate exactly the same p73 staining pattern in skin. Chemicon rabbit polyclonal AB7824 [4], Leica Biosystems mouse monoclonal (Leica Biosystems Cat# NCL-p73, RRID:AB_563940), and two Abcam’s: EP436Y (aa 50–150) [42,48], EPR19884 (C-terminus, skin IHC image is at the Abcam website). Reactivity of the Abcam’s antibodies to p73 is validated by the knockout/knockdown and overexpression experiments either by manufacturer or by independent investigators using immunohistochemistry and immunoblotting [42,60]. The main concern is that p73 antibodies can cross react with p63. There are several lines of evidence against it. Firstly, p63 immunostaining appeared earlier during skin development than EP436Y p73 staining [42]. Secondly, during oocyte development, EP436Y p73 antibody stains granulosa cells and these staining disappear in the p73 −/− mice. In contrast, p63 stains primordial follicles in both p73+/+ and p73 −/− mice [60]. These data were supported by immunoblotting of the wild type and p73 −/− ovaries. Third, the p73 EP436Y antibody does not react with p63 by immunohistochemistry [48] as it stains a subset of the p63 positive cells in human and mouse epidermis [42,48]. Moreover, closer examination revealed that a few cells have brighter p73 staining than p63 [42,48]. The absence of reactivity of the EP436Y p73 antibody to p63 by immunoblotting was also demonstrated [42]. Therefore, it is possible to conclude that at least EP436Y p73 antibody is reliable.

In addition, several tissues display strong p63 staining and no AB7824 or RRID:AB_563940 p73 antibody staining (breast, prostate, placenta, seminal vesicles) was observed, whereas other epithelial organs display the same staining of the basal cells as in skin (Figure 2) http://www.proteinatlas.org) [50].

Thus, p73 protein is expressed in the basal layer of epithelial organs, whereas p63 appears in the basal as well as in the differentiated cells.

### 3.2. Common and Distinct Multiprotein Complexes of p53 Family Members and Their Distribution in the Epithelial Organs

Differential and common functions of p63 and p73 are reflected by the proteins they interact with. In order to have the available information of their binding partners, we downloaded protein-protein interactions for the p53, p63 and p73 from the BioGrid database. We excluded interactions from the genetic screens, keeping only physical associations. Shared and unique interactions are presented as an Euler diagram (Figure 3A).

Furthermore, we annotated shared and unique p53, p63 and p73 interactions using Metascape software [53]. The MCODE algorithm within Metascape detects densely interconnected regions in protein-protein interaction networks that may represent distinct molecular complexes (Figure 3B–D). In addition, Metascape applies enrichment analysis of pathways and processes to identified networks of protein interactions and uses the top three enriched terms for the annotation of biological roles of the putative molecular complexes. In addition, we compared protein interaction data with mRNA expression using publicly available TCGA RNA-seq datasets and single cell sequencing of human skin (Appendix A) [55]. Below we review what is known about the functional impact of the p53 family protein interactions focusing on the biology of the epithelial tissue.

#### 3.2.1. Interaction between p63 and p73 in Gene Regulation

It is widely accepted that transcription is regulated by the interplay of the multiprotein complexes [61,62,63,64,65]. Moreover, due to the presence of dominant activators or repressors co-occupying DNA regulatory sites, functions of a particular transcription factor might be limited to a subset of its binding sites [61,64,66,67,68]. For the p53 transcription factors family, interactions with distinct proteins may skew the binding of p53, p63 and p73 isoforms towards different sites in the genome and determine their specific activities [14,21,43,63,69,70,71].

As discussed above, p63 and p73 colocalize in the basal layers of the epithelial organs, while p73 is not present in the differentiating cells. Indeed, the p63 and p73 can hetero-oligomerize through their tetramerization domains, and the resulting hetero-tetramer consisting of two p63 and two p73 molecules is a thermodynamically preferred structure [48].

Calcium differentiation of the keratinocytes induces p73 protein and promotes its interaction with p63 at days 2–4 of calcium-induced differentiation [48]. The interaction between p63 and p73 is initiated just before cell culture stratification and might play a role in the induction of differentiation markers. However, it is important to mention that at 48 h in calcium-enriched media, cultures of human keratinocytes typically contain a mixture of proliferating and differentiated cells, thereby, it is possible that interaction between p63 and p73 occurs in the proliferating cells, consistent with in vivo p73/p63 colocalization in the basal cells.

In support of this data, it was shown that in HaCat and NHEK cells p73 delta and gamma isoforms as well as p63 were able to activate involucrin and loricrin promoters suggesting that both of these genes may play a role in the induction of differentiation [72].

Interactions of DNp63a and TAp73β led to reduced TAp73β transactivation of the luciferase reporter in neuroblastoma cells [48]. However, in skin and other epithelia p73 and p63 are mostly represented by the DN- isoforms, whose transcriptional activation is determined by interaction with other transcription factors [14,21,71]. The distinction between DN- isoforms of p63 and p73 is demonstrated by co-expression of the KLF4 with DNp73β (but not DNp63α or either KLF4 or DNp73β alone) that dramatically induced fibroblast–keratinocytes reprogramming and expression of keratinocyte specific marks (KRT5, KRT14, FLG, SPN) in both fibroblasts and mesenchymal triple-negative breast cancer cell line MDA-MB-231 [42].

Several signaling transcription factors/pathways cooperate with the p63/p73 to regulate gene expression. Of these, the most studied are CEBP/AP1/NfKb transcription factors that play a pivotal role in the epithelial differentiation [61,73].

It is likely that in skin cells DNp63a and DNp73α interact with either c-Jun or c-Rel and localise to the AP-1 binding sites similarly to what was observed in neck squamous cell carcinoma and osteosarcoma Saos cells [14,21]. In the squamous cancer cells, proinflammatory cytokine TNF-α modulates interaction of DNp63α/TAp73 with c-REL and induces redistribution of TAp73 from the p53 to AP-1 DNA binding sites repressing TAp73 pro-apoptotic functions and inducing an oncogenic gene expression in the absence of functional p53 [21,71]. Thus, TAp73 in complex with AP-1 versus DNp63a switch from anti- to pro-oncogenic activities [15,21,71].

Accordingly, a consensus p53 binding motif is found next to the composite CEBP/c-Jun binding site enriched in the promoters bound by CEBPb and c-Jun in mice basal keratinocytes [61]. Notably, promoters with differentiation induced RNA polymerase 2 binding are preferentially occupied by CEBPb and c-Jun. However, the combinatorial p53/CEBP/AP1 motif was overrepresented in the CEBPb bound promoters also active in the basal keratinocytes.

In addition, it was shown that p63 interacts with CEBPb and Nf-kb in lung cancer cells upon cigarette smoke extract exposure and regulates COX-2 expression, supporting possible functional consequence of the p53/CEBP/AP1 binding sites proximity [74].

The localization of the p63 binding sites as suggested by the ChIP-seq data does not change upon the keratinocytes differentiation and modulation of the activity of the p63-bound enhancers is likely determined by other protein complexes that cooperate with p63 [66]. Upon progression through the cell cycle during epidermal differentiation, high expression of the chromatin remodelers occurs in the mitotic phase of the cell cycle, concomitant with induction of the early differentiation markers KRT1 and KRT10 [55,57], possibly marking a subset of epidermal asymmetrical division, in which p63 plays a pivotal role [35]. Accordingly, p73 interacts with chromatin remodelers and transcription factors (Jun, RB1, SP1, HDAC1) involved in the cell cycle regulation, FOXM1 pathway and mitosis (Figure 3), suggesting that the p73 specific interactions might modulate p63 activity either by promoting or repressing transcription [15,63,75,76,77].

Indeed, additional evidence from the chromatin immunoprecipitation sequencing (ChIP-seq) data suggests that p63 and p73 occupy a similar set of promoters in cancer cell lines [42,78]. Even though genome-wide colocalization of the p63 and p73 in the primary keratinocytes or in skin has not been yet analyzed, it is assumed that p63 and p73 will be localized similarly [42]. Thus, both hetero-tetramerization mediated and distinct DNp63 and DNp73 DNA binding are likely to occur in the epithelial organs. While a plausible hypothesis would be that these complexes differentially regulate transcription in the basal epithelial and in the differentiating cells, details of the regulation are just beginning to emerge [42,48,66,74].

#### 3.2.2. Common p63 and p73 Interactors

Complexes of 52 proteins that interact with both p63 and p73 correspond to the shared pathways regulating the p53 family. These include regulators of the p53 stability and transcriptional activation by ubiquitination and sumoylation: MDM2, MDM4, RCHY1, ITCH, SUMO1, UBE2I [22,79,80,81,82,83,84,85], transcriptional co-regulators: HDAC2, EP300, KAT2B, HIPK2 [77,86,87,88,89,90], DNA repair gene BRCA1 that interacts with p63 [91] while repressing TAp73 expression and chemoresistance [92], DNA binding transcription factors: HIF1a, YAP, ATF3, p63, p73 [93,94,95]. Two proteins that are bound by p63 and p73 and not p53 are CDC20—a pivotal protein in the mitosis [96,97] and splicing regulating protein FUS [98,99].

It was shown that HDAC1 and HDAC2 depletion phenocopy p63 knockout in mice derepressing negatively regulated DNp63 targets CDKN1A(p21), SFN(14-3-3σ), and CDKN2A [37,100]. Interestingly, classical p53 target and cell cycle inhibitor CDKN1A [101] is expressed strictly in the suprabasal cells in human epithelial organs, especially in the esophagus and vagina. A p53/p73/CDKN1A inhibitor MDM2 is also in this group with unrestricted expression in all skin cells [22,84,102].

#### 3.2.3. p73 Interactions

Analysis of 89 p73 or p53 interacting proteins revealed two putative protein complexes involved in the cell-cycle progression, chromatin modification complex and adaptive immunity complex (Figure 3C).

Analysis of 36 p73-only interacting proteins revealed clusters of interactors pointing to their potential role in the mitotic prometaphase, and resolution of the sister chromatin cohesion and segregation. These include protein of the spindle assembly checkpoint complex (Figure 3C, complex 1) (CCNB1, MAD2L1, Bub1B, STAG1, DSN1) [96,103,104,105,106,107,108].

In turn, 53 p73 interacting proteins that are also known to bind p53 but not p63, represent histone modifiers including: histone deacetylases: HDAC1, CHD3 [85,109], histone acetylases: KAT5, CREBBP [110,111], H3K4me2 or H3K9me2 lysine demethylase KDM1A [112]. The p73 and p53 interact with proteins involved in G1/S transition of the cell cycle (SP1, MYC, CCND1, RB1) [76,106,113,114,115] as well as proteins involved in the G2 to M transition Cyclin A and CDK1 [106,116]. Also, p73 and p53 interact with Aurora A kinase that represses p73 mediated transcriptional activation and SAC functions [96,117,118].

This group includes a classical p53/p73 target p21 (CDKN1A)—a major regulator of the G1/S cell cycle progression that acts by dephosphorylation and repression of multiple proteins involved in the cell cycle progression [101]. Accordingly its expression is restricted to the suprabasal layer of skin, vagina and other epithelial organs with no proliferation [101,119]. However, p21 protein does not interact with p73 directly, only p73 binding to the p21 promoter has been described, although p21 interacts with many proteins that do interact with p73, including SP1, RB1, CyclinD1 and CyclinA2.

Analysis of 58 p53/p73 interacting proteins that are highly expressed in the skin [55] revealed putative protein complexes involved in the cell cycle regulation and FOXM1 pathway: CCND1, CCNA2, CCNB1, CDK1, CDK2, Rb, transcriptional regulation: MYC, HDAC1, KAT5, CREBBP, cellular response to stress: HSPA4, HSPA9, NFYB, microtubule organizing center and G2/M transition:PLK1, AURKA.

#### 3.2.4. p63 Interactors

Gene annotation of 187 p63-only interacting proteins reveals significant fraction of them being involved in RNA splicing and TGFb response. Chrondate and nervous system developmental pathways are highly represented in this group as well.

The TAp73 interacting with APC/C(CDC20) repress progression to anaphase and aberrant chromosome segregation, while APC/C(CDC20) interaction with DNp63a promotes DNp63a degradation and is required for keratinocytes differentiation [97]. Notably, p63 interacting APC/C components CDC20 and ANAPC2 are located strictly in a few cells of the basal and suprabasal layer, likely marking the last division of the cells committed to differentiation.

However, in the suprabasal layer, after keratinocytes differentiate, p63 protein is abundant suggesting that DNp63a isoform that is sensitive to the APC/C (CDC20, CDH1) mediated degradation is more restricted to the basal keratinocytes, whereas DNp63 present in the differentiated layers is likely other isoforms such as DNp63b and DNp63g [97].

Interestingly, one of the p63 interacting proteins is chaperonin containing TCP1, subunit 8 (theta) (CCT8) [98], that according to high throughput screening in turn interacts with gamma tubulin1 [120,121,122] and component of SAC complex BUB1 [123], suggesting yet unexplored implication of the p63 in the spindle checkpoint that might interact with or compete with p73.

Among proteins interacting with p63 are WWP1, C/EBPb, SMAD4 and KLF5. The WWP1 is an E3 ubiquitin ligase that is required for p63 transcriptional activity and WWP1 depletion causes cell cycle arrest [124]. Consistently, WWP1 displays an interesting staining pattern in skin, being nuclear in the basal cells and strictly peri-nuclear in the differentiated keratinocytes. Moreover, WWP1 ubiquitinates and leads to degradation and repression of KLF5 [125,126] that is highly expressed in the nuclei and cytoplasm of the basal and differentiated keratinocytes and of SMAD4 [127] that is localised to the cytoplasm in the epithelial tissue.

C/EBPb and C/EBPa are pivotal factors in the induction of genes during keratinocytes differentiation [61,73]. While, interaction between C/EBP and DNp63 leads to the induction of COX-2 expression in the lung epithelial cells upon smoke extract exposure [74], COX-2 induction in the skin keratinocytes leads to PGE2 induction and inflammation in response to ultraviolet irradiation [128]. Therefore, the functions of WWP1 interacting proteins is this subgroup are likely to be a switch of transcriptional regulation from basal to differentiated program.

Another protein interacting with p63 and WWP1 is KLF5. Both KLF5 and KlF4—a well-known pluripotency Yamanaka factor [129]—are expressed in skin with distinct staining patterns. The KLF5 decorates nuclear borders just like WWP1, whereas KLF4 is highly expressed throughout the differentiated cells. KLF4 is not known to interact directly with p63, however, it was shown that p63 regulates KLF4 expression and combined p63/KLF4 expression differentiate fibroblasts into keratinocytes [130,131]. In contrast, the role of p73 in reprogramming is controversial, because while one study used p73 depletion technique to demonstrate that p73 has no role in reprogramming [131], the other concluded that overexpression of DNp73β and KLF4 leads to reprogramming [42].

Common p63 and p53 interacting proteins relate to autophagy and protein folding (CHEK2, RELA, HSP90AB1) Figure 3D. Among p63 interacting proteins are several 60S ribosomal subunits including RPLP0, RPL3, RPL12, RPL13A, RPL24 as well as 40S ribosomal protein SA and UPF2 that controls nonsense-mediated mRNA decay [98]. In addition, p63 binds several human ribonucleoproteins HNRNPK, HNRNPL HNRNPA3, HNRNPR [98], HNRNPD [132] and HNRNPAB [133] suggesting that p63 isoforms function in the splicing regulation (Figure 3D).

#### 3.2.5. p53 Interactions

It was not possible to annotate 971 unique p53 interactions by the Metascape. When we limited our analysis to 604 proteins that are detected in skin by the single cell RNA-seq [55], the Metascape recognized 11 putative protein complexes with functions ranging from the DNA repair to mRNA splicing and metabolism (Appendix A) consistently with known p53 functions in the genotoxic stress responses, transcription and RNA regulation pathways [121,134,135,136].

### 3.3. Distinct Roles of p63 and p73 in Epithelial Tissue

#### 3.3.1. The Role of p73 in Mitosis and Regulation of Polyploidy

One of the p73 functions that explain its preferential localization to the basal cells of epithelial tissue is the regulation of cell ploidy. A role of the p73 in mitosis has been described [103,106,116,137,138,139,140]. Accordingly, p53−/− p73−/− primary mouse embryonic fibroblasts become polyploid after three passages, whereas a majority of p53−/− MEFs were diploid and polyploid cells appeared only after passage 8 [23].

As discussed above, p73 interacts with components of sister chromatids cohesion and segregation such as cohesin component STAG1 that provides sister chromatid cohesion along the length of a chromosome from DNA replication through prophase until prometaphase [141,142]. Notably, STAG1 is localized to the basal and suprabasal layers of skin and other epithelial organs where the last division occurs.

Also, p73 interacts with DSN1, a component of the kinetochore hMis12 complex that links Bub1b to the kinetochore [140,143,144].

Phosphorylation of the p73 S235 by Aurora-A kinase results in the inactivation of its DNA damage and spindle assembly checkpoint response functions and mitotic exit is accelerated due to release of the Mad2-Cdc20 spindle assembly checkpoint complex [96,140]. p73 phosphorylated by Aurora-A loses the chromatin binding affinity, and becomes sequestered in the cytoplasm [96], thereby halting the expression of its targets such as p21 [96] and pro-apoptotic BH3-only protein Bim and hence affecting cytochrome C release and caspase activation [138].

In lung adenocarcinoma cell line H1299 expression of the DNp73β isoform of p73 leads to tetraploidy and cell death through mitotic catastrophe rather than apoptosis [139]. In contrast, in the osteosarcoma cell line Saos2, induction of ectopic p53, ΔNp73α, and TAp73β or γ had no effect on the number of polyploid cells, while TAp73α overexpression led to a significant increase in the number of polyploid cells [145]. Despite authors claiming that DNp73α has no effect on the polyploidy, in Saos cells expressing DNp73α the fraction of polyploid cells was 50% compared to the control cells at 24 h and about 30% in relation to control after 48 h of DNp73α induction. Mechanistically the effect was linked to the interaction of Tap73α with Bub1 and Bub3 that were also overexpressed in these cells. Notably, neither p53 nor any of the other p73 isoforms interacted with Bub1 and Bub3. Notably, in glioblastoma cells, overexpression of the DNp73 isoform was associated with the occurrence of an abnormal number of centrosomes, while ectopically Tap73 overexpressing cells showed normal centromers count with no association with BubR1 [105].

Consistently with these results, in the SW480 and MDA-MD231 cell lines, TAp73 isoform interacts with BubR1 whereas DNp73 does not [103]. However, pan-reactive antibody staining revealed that TAp73 protein level was about 4x higher than DNp73 in these cell lines. TAp73 C-terminal domain that is a part of the DNp73 isoform was responsible for the interaction. Moreover, when DNp73α, DNp73β, TAp73α, and TAp73β were overexpressed in HeLa cells, they bound exogenous BubR1 at a similar level. However, only TAp73 supported p-H3 phosphorylation and while DNp73β did not, and DNp73α was not tested. Different effects of ectopically expressed p73 isoforms on polyploidy—in Saos2 with DNp73α [145] and in H1299 with DNp73β [139]—may correspond to the SAM domain or/and to cell line specific factors.

In the basal keratinocytes in skin approximately 75% are in the DNp73α isoform and 20% are in the DNp73β. Hence, our hypothesis that in the basal epithelial cells DNp73 can control polyploidy does not necessarily contradict these experiments. However, whereas the role of TAp73 in the SAC is well established, the role of DNp73 isoforms is not clear yet and demands further investigation.

#### 3.3.2. The Role of p63 in Transcription-Coupled Splicing

The DNp63 [146,147,148] as well as alternative splicing [149] and polyadenylation [150] play a major role in keratinocytes differentiation and in cancerogenesis [151].

Several proteins uniquely interact with p63, specifically with DNp63a containing SAM domain in keratinocytes bind and regulate RNA metabolism (Figure 3C) [98].

In contrast to the APC/C components, p63 interacting ribonucleoproteins are found in all cells of the basal and differentiated epithelial layers. Consistent with continued presence of the p63 in the differentiated keratinocytes, it is possible to speculate that these interactions regulate differentiation-specific RNA splicing. For example, physical association of the heterogeneous nuclear ribonucleoprotein A/B (HNRNPAB) and p63a via its SAM domain led to a specific shift of FGFR-2 alternative splicing towards the K-SAM isoform essential for epithelial differentiation [133]. In addition, DNp63a indirectly promotes TERT expression but shifts mTERT alternative splicing towards dominant negative isoform by interacting with HNRNPAB [152]. Thereby, DNp63a mediated splicing might be a component of a negative feedback loop and overall effect of DNp63a overexpression sums up to the mTERT activity downregulation resulting in the induced senescence of mouse epidermal keratinocytes [152].

Another case of the p63 mediated RNA regulation is p63 interaction with DHX9 (RNA helicase A) that is highly expressed in differentiated keratinocytes and is known to regulate circRNA formation through binding to the inverted-repeat Alu elements [153]. The circRNA expression is induced in the differentiated keratinocytes [154] and the role of p63 is yet to be determined.

In addition, the DNp63a interacts with RNA binding protein FUS [98], that is required for transcription couples splicing [155]. FUS functions in coupling transcription to splicing via mediating an interaction between RNAP II and U1 snRNP [155]. Acute depletion of U1 snRNA or of the U1 snRNP protein component SNRNP70 markedly reduces the chromatin association of hundreds of lncRNAs and unstable transcripts, without altering the overall transcription rate in cells [156]. It was shown that upon DNA damage in squamous cell carcinoma cells, ATM kinase is a master switch for the DNp63α phosphorylation at S385 and it’s degradation upon subsequent phosphorylation by CDK2 and p70s6K kinases [157]. Subsequently, it was shown that S385 DNp63a phosphorylation is induced by cisplatin in squamous cell carcinoma cells, leading to RNA splicing and ACIN1-mediated cell death via interaction with spliceosome components SAP18, RBM38, ELAVL, SRPK2, SRSF2, and ACIN1 [158]. The p63 mutations within the SAM domain cause ankyloblepharon, ectodermal dysplasia and clefting [133,159]. The p63 carrying mutation in the SAM domain modulates its own splicing leading to accumulation of the C-terminal truncated isoform of the p63 resistant to proteasomal degradation [159].

Thus, direct and indirect evidence suggest that p63 mediated regulation of RNA metabolism influences basal and differentiation specific programs in epidermis [98,133,152,156].

The role of p73 in promoting translational elongation is recently revealed [160], although, the mechanisms are largely unknown and the effect on translational elongation and protein synthesis was mimicked by the depletion of two p73-regulated transcripts—XPO1 and UTP18 [160]. MYCN plays an important role in neuroblastoma pathogenesis, and it was discovered that TAp73α could interact with and destabilize MYCN mRNA in glioblastoma cells [161]. Other examples of p73-mediated post-transcriptional RNA metabolisms or translation are yet to be discovered.

Thus, direct and indirect evidence suggest that p63-mediated regulation of RNA metabolism influences basal and differentiation specific programs in the epidermis [98,133,152,156].

Finally, we address a question if distinct p63 and p73 protein interactions correspond to conserved or diverse protein domains (Figure 4).

While p63 binding sites for the p63 interacting complexes are not mapped, a lot more is known about putative p73 interacting complexes (Appendix A, page “p73 complexes and annot”). We plot mapped interacting pieces of p73 or amino acids of the p73 along the domain structure and frequency of conserved amino acids in between p63 and p73. Apparently, the amino acid sequences are not conserved except DNA binding domain and 9 out of 10 protein-binding regions are within divergent transactivation, oligomerization or SAM domains. However, 4 out of 6 protein binding and phosphorylation sites are conserved in between p63 and p73, which is consistent with 13 out of 23 p73′s post-translational modifications (PTM) from the dbPTM database being conserved in the p63 [56] (Figure 4B). Notably, Tyr99 phosphorylation site of the p73 corresponds to phosphorylation of p63 within conserved stretch of amino acids at Tyr 149 by ABL1 kinase [162], suggesting that some interactions from the BioGrid database need to be updated or additional interactions will be discovered in the future. In addition, Tyr487 ubiquitination of p73 by NEDD4 [163] corresponds to Tyr542 ubiquitination of p63 by the WWP1–E3 ubiquitin ligase of the same family [164]. Thus, distinct p73 associated complexes bind divergent domains, however, critical protein kinase recognition sites that regulate these binding events are conserved.

## 4. Discussion

Aberrant expression of the p53 family members protein isoforms underline progression of many cancers by regulating the balance between proliferation and apoptosis [165,166,167,168,169,170]. Here, we focused mostly on p63 and p73, whose activity contributes to pro- and anti- apoptotic signaling pathways and the spindle assembly checkpoint complex and mitotic exit. Abnormalities in this pathway are implicated in cancer progression and cell polyploidy. In normal epithelial organs, p73 is expressed strictly in the basal cells, whereas p63 is expressed in both basal and suprabasal cells committed to differentiation and p53 is not detected. Notably, polyploidy is known to occur in the differentiated epithelial cells, suggesting that the absence of p73 might contribute to the polyploidy/multinucleation. We propose that in the normal epithelial organs, p73 also regulates polyploidy and multinucleation of non-transformed cells. Alternatively, p73 is redundant in the regulation of polyploidy and its functions are limited by the regulation of the basal cell functions.

First, cluster analysis of the protein interactions by the Metascape obtained from the open source database (Biogrid) highlights a unique role of p73 in the regulation of cell cycle and mitosis and, unexpectedly, suggests a unique role of p63 in the RNA metabolism and related cellular functions. Moreover, literature suggests that a specific p73 isoform, DNp73β, plays a dominant and yet redundant role in the basal keratinocytes differentiation and related functions, although the effects on polyploidy are yet to be analyzed.

## 5. Perspectives and Future Directions

There are unresolved questions that need to be answered. While a role of the TAp73α isoform in the mitotic checkpoint is established, the role of the DNp73α isoform which is expressed in the basal keratinocytes of skin is not clear yet. Other proteins, central to SAC regulation, are expressed in the skin but details of the SAC checkpoint control which is established mostly in the cancer cell lines is likely different in the normal epithelial cells.

Surprisingly, transcriptional and post-transcriptional regulation of p63 and p73 via protein–protein interactions are not well understood. Apparently, p63 and p73 isoforms regulate chromatin modifications in a cell specific and context dependent manner. Moreover, the issue is complicated by the fact that p63/p73 isoforms bind to DNA not only as transcription factors, but also as a non-DNA bound component transactivating other transcription factors (Jun, c-Rel, NFYB). Few investigations suggest that differential activity of p63/p73 is complex with each other versus other p63/p73 specific interacting proteins switch TAp73 from anti- to pro-oncogenic activities [15,21,71]. These data are prompting researchers to examine if DNp73 can be switched from pro- to anti-oncogenic activities considering that DNp73 is overexpressed in many cancers and DNp73 can be an activator when interacting with other transcription factors [169,171,172]. Taking into account the p63 interactions with several hnRNPs playing a dual function in transcription and translation [173], it is possible to speculate that the p63/p73 complex is involved in the RNA biogenesis. Therefore, identification of the specific p63/p73 transactivation domains and their modifications that drive chromatin modifications and regulate gene expression and splicing will be a hot topic of future investigations.

## Figures and Tables

**Figure 1 cells-10-00025-f001:**
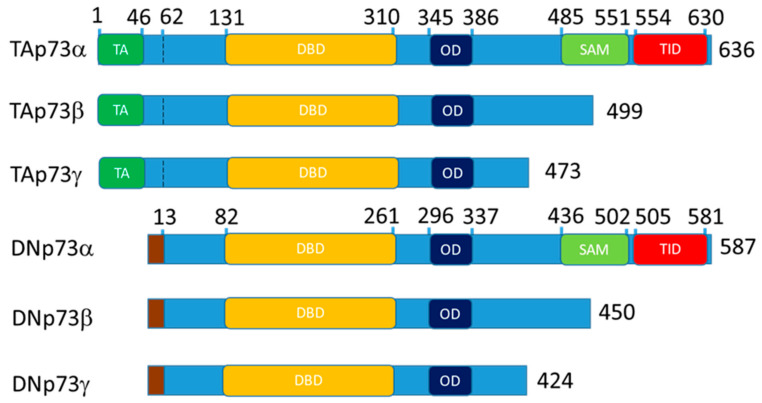
Schematic representation of the p73 domain structure for the main isoform expressed in the epithelial tissue. The p63 follows the same nomenclature. Altogether, 13 isoform of p63 and 13 isoforms of p73 exist. DN- isoforms begin with the stretch of 13 amino acids that replace TA domain at 62 bp. The TAp73a coordinates are for the O15350 protein and DNp73a coordinates are for the BAB87244.1 protein. TA—transactivation domain, DBD—DNA binding domain, OD—oligomerization domain, SAM—sterile alpha motif domain and TID—transcription inhibitory domain.

**Figure 2 cells-10-00025-f002:**
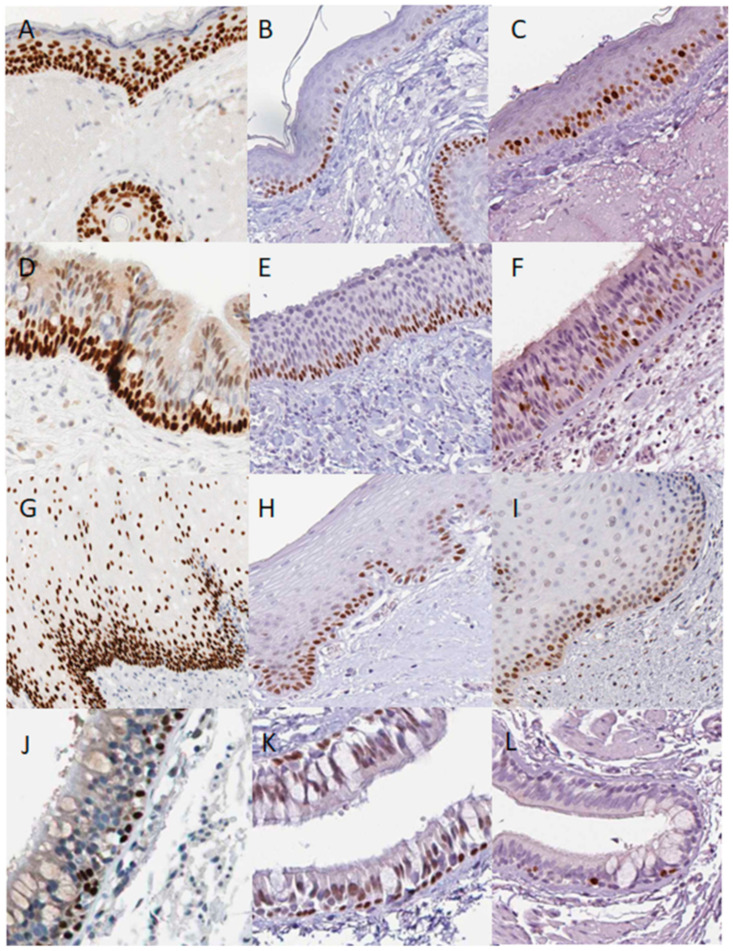
Distribution of p63, p73 and mitotic marker MKI67 in normal human epithelial tissues according to the Protein Atlas database. Skin p63 (**A**), p73 (**B**) and MKI67(**C**), Nasopharynx p63 p63 (**D**), p73 (**E**) and MKI67 (**F**) staining; oral mucosa p63 (**G**), p73 (**H**) and MKI67 (**I**) staining; lungs bronchial epithelium p63 (**J**), p73 (**K**) and MKI67 (**L**) staining. In normal human epithelial tissues, the p73 antibody stains basal cells, whereas p63 antibody stains basal as well as early differentiating cells. Proliferating MKI67 positive cells are detected in the basal and suprabasal layers of the tissues, except the non-proliferative bronchs.

**Figure 3 cells-10-00025-f003:**
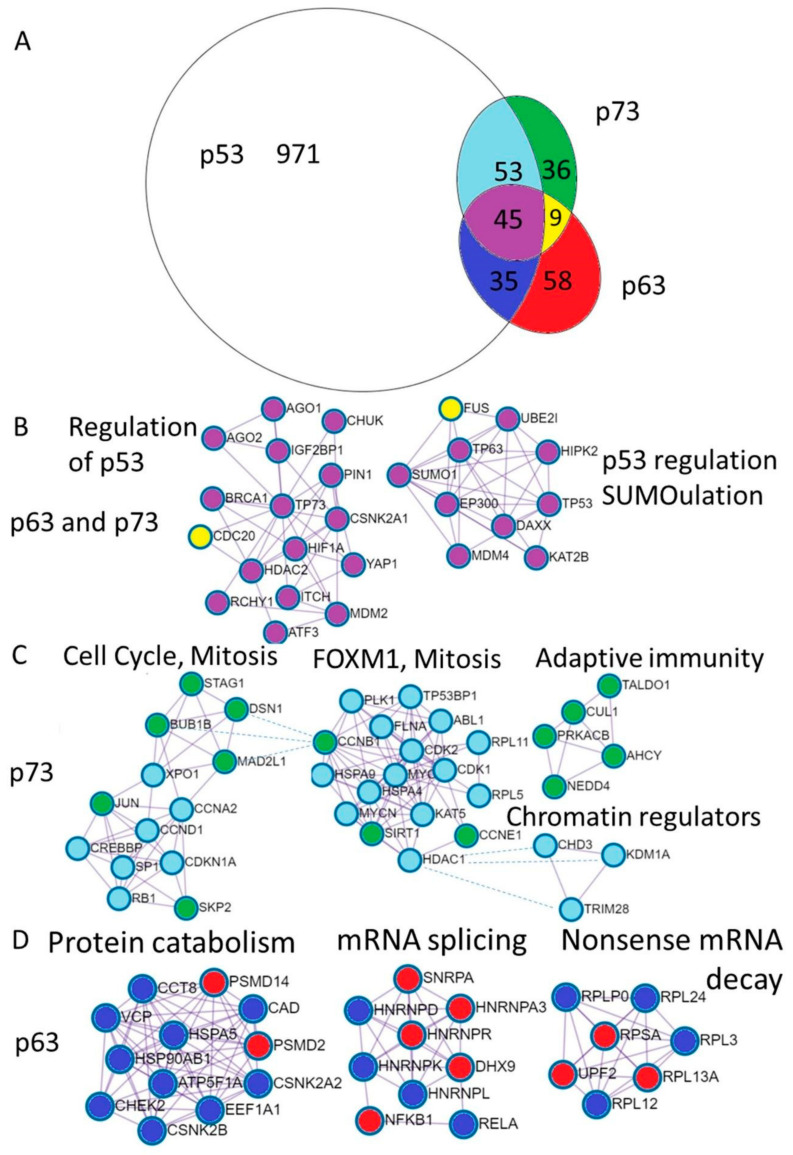
Annotation of the unique p73 and p63 protein interactions reveal distinct physiological functions. (**A**). Euler diagram of the p53, p63, and p73 protein interactions from the BioGrid database. Coloring of the Euler diagram corresponds to the coloring of dots representing proteins in the subsequent sections of the figure (**B**–**D**) describing protein interactions. (**B**). Annotation of proteins uniquely interacting with p63 revealed putative complexes involved in the protein catabolism and synthesis including protein folding and transport (CCT8, HSPA5, HSP90AB, VCP, CCT8, ATP5F1A), translation—EEF1A1 and degradation (PSMD2, PSMD14). The second complex contains ribonucleoproteins involved in the mRNA splicing and Nf-kb transcription factors and the third represents ribosomal proteins involved in translation and nonsense mRNA decay (UPF2). (**C**). Annotation of proteins interacting with p73 revealed putative complexes involved in the regulation of the cell cycle (Rb, CDKN1A (p21), Cyclin D1, Cyclin A2, SPK2) as well as mitosis (BUB1B, MAD2L1, DSN1, STAG1). The next complex is FOXM1 mediated cell-cycle regulation. Interestingly, the adaptive immunity cluster is composed exclusively of the p73 interacting proteins and three chromatin regulators can bind both p53 and p73. (**D**). Annotation of proteins interacting with both p63 and p73 revealed putative complexes involved in the regulation (repression) of the p53 stability (MDM2, MDM4,RCHY1, ITCH, SUMO1, UBE2I) and transcriptional activity (HDAC2) transcriptional coactivators (EP300, KAT2B, HIPK2), DNA repair (BRCA1), DNA binding transcription factors (HIF1a, YAP, ATF3, p63, p73). Two proteins that are bound by p63 and p73 and not p53 are CDC20—a pivotal protein in the mitosis and FUS—the splicing regulating protein.

**Figure 4 cells-10-00025-f004:**
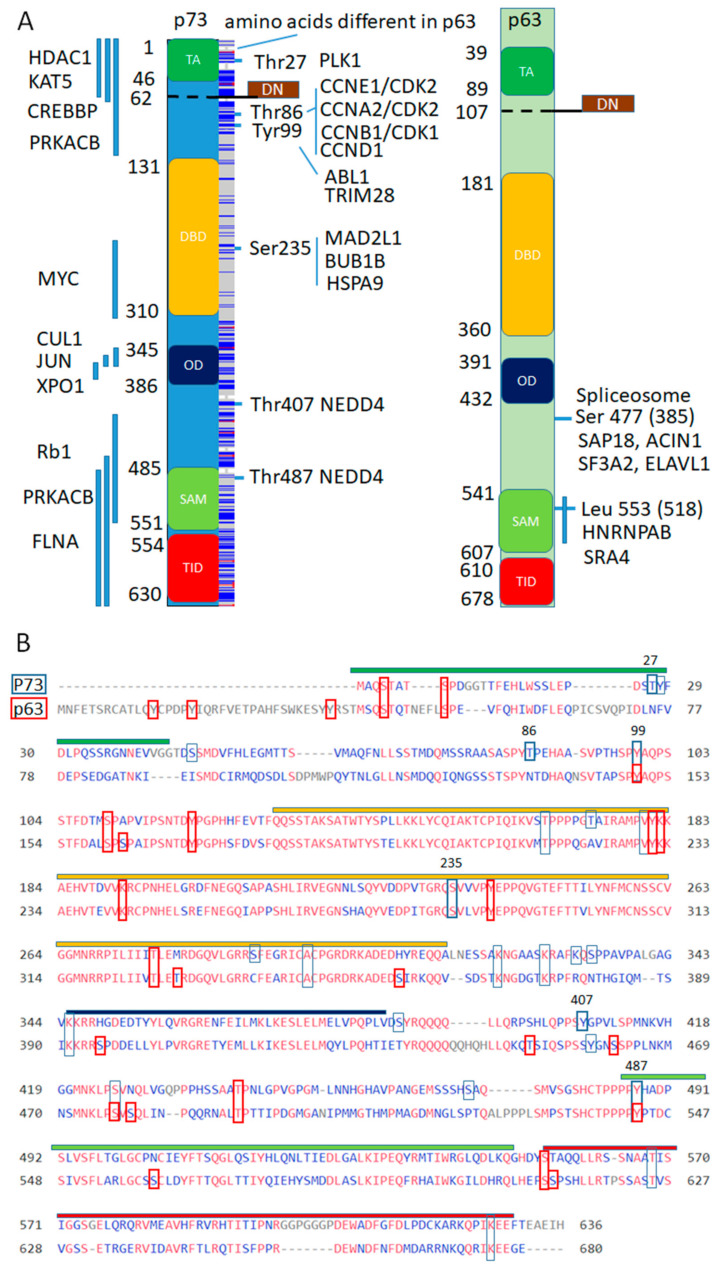
Conservation of interaction sites and domains between p63 and p73 for putative protein complexes interacting with p73 and not with p63. (**A**). Domain structure of p73 and p63, with highlighted domains along with mapped interacting regions on the left shown as vertical blue lines and specific amino acids on the right with corresponding interacting proteins. Blue lines in the box along the p73 domain structure represent amino acids that are different in the p63. DN- domain is a result of the alternative splicing that replaces TA- domain at the N-terminus. (**B**). Alignment of p73 (O15350) and p63 (Q9H3D4) sequences with perfectly aligned shown in red. In blue quadrants—p73 post-translational modifications and in red—p63 post-translational modifications (PTM) from the dbPTM (http://dbptm.mbc.nctu.edu.tw) with numbers marking amino acids that participate in distinct p73 protein interactions. Conserved PTM sites in between p63 and p73 are marked by elongated quadrants.

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
