# Peer review of "Distinct p63 and p73 Protein Interactions Predict Specific Functions in mRNA Splicing and Polyploidy Control in Epithelia"

_cells, 2020, doi:10.3390/cells10010025_

Round 1
Reviewer 1 Report
Rozenberg et al. use a range of published resources to review the potential roles of p63 and p73 in the human body. They make good use of histology, interaction data, and literature to describe these proteins and their potential roles. The authors should however be careful to use (and describe) their approaches to identifying interacting proteins and the statistical enrichments in these interactions. Generally, the authors have produced a nice review of these protein functions although I have a number of specific concerns/suggestions that should be addressed.
Comments
- The authors suggest that p53 is poorly expressed at the mRNA and protein level in most tissues (128-132). While this is undoubtably true of the protein, p53 mRNA is actually quite strongly expressed in many tissues (10-40 TPM). The authors should carefully compare the mRNA expression levels of these three proteins in addition to the protein levels. It is also worth noting that p53 is a stress response factor and its abundance could change greatly depending on the circumstances of the tissue (eg. during injury, or toxic insult).
- When quoting numbers from external sources the authors should be careful to give units. For example lines 160-164, 7.8 what? It is also unclear what the authors are referring to when they note “this treatment” can lead to RNA degradation.
- If the authors are relying on published data to validate antibodies they should do a more complete literature review. Focusing on one or two of the antibodies and carefully describing the papers that reference these antibodies. Their current discussion of the problem seems somewhat circular and I am not confident in their strong conclusion that the EP436Y antibody is reliable.
- In the section starting at line 223 the authors discuss gene regulation by p63/p73. Though generally reasonable the authors could ground their discussion better. For example, the authors could note that the three proteins all bind to an essentially identical DNA motif. Further with regard to homo/hetro-tetramerization the authors could describe the potential consequences or role of this distinction. Finally, the authors would be advised to streamline this section, as I am not sure what the take-away message is? P63/p73 interact with various transcription factors on and off chromatin in an isoform specific manner?
- The section on common p63/p73 interactions (273-286) is somewhat interesting, but also difficult to conclude anything from. Are we to conclude p63/p73 has similar or different interactomes? The final sentence on negative feedback is difficult to make sense of as physical interactions do not clearly relate to negative feedback control.
- For the section on each specific proteins interactions (287-364) some statistics are badly needed. Are these interactions significantly enriched? With what pvalue. In what datasets?
- The discussion of p73 in polyploidy is well balanced
- The final section on p63/p73 alignment is nice, but the figure could perhaps be clearer. Perhaps the left panel could be expanded to show both proteins, their conservation, domains, and putative interaction regions. This would likely be more informative than the amino-acid based comparison in the right panel.
- The discussion is well written and reasonable. A valuable addition would be a description of the potential roles of p63/p73 in cancer cells.
Reviewer 2 Report
This manuscript focused on p63 and p73 that are two members of the p53 family. The authors of this manuscripts started to show that in epithelial tissues p73 is expressed specifically in basal cells, whereas p63 is presents in both basal and differentiated cells. Next, by analyzing protein-protein interaction networks, they suggest that these proteins could also have different functions. In particular, they showed that p73 interacts with factors involved in transcriptional repression, cell cycle and mitosis, whereas p63 interactors play roles in the regulation of RNA metabolism (in particular splicing).
This is a well written and informative work that will certainly provide a useful reference for further studies.
I suggest only minor changes:
This review could be further benefit through the inclusion of an additional section, such as a critical and well organized "Perspectives and Future Directions" paragraph.
Author Response
Dear Reviewer.
Thank you very much for reading our manuscript.
We incorporated requested section at the end of the manuscript.
Sincerely.
Julian Rozenberg, PhD.
Reviewer 3 Report
In the present review, by an exhaustive and elegant way, the authors reported the most relevant roles of the p63 and p73 transcriptional factors in specific pathways regulating epithelial cell proliferation and differentiation. The authors used genomic and proteomic tools to dissect the interactions of p63 and p73 with other factors and co-factors. From these evidences they have extrapolated and validated from the literature the main pathways governed by these two members of the p53 family.
Interestingly they found that p73 and p63 cooperate in the maintenance of genomic stability, in the control of the cell cycle checkpoints (anaphase promoting complex and spindle assembly checkpoint), whereas p63 interacting with specific interactors is able to regulate mRNA processing and splicing in both proliferating and differentiated cells.
The knowledge of the mechanisms modulated by p73 and p63 and how they intercalate with pro-proliferative and pro-differentiative signaling pathways is an excellent starting point for targeted molecular therapy strategies of skin disease.
Overall, this review was well written and it well addressed the main topic.
Minor point:
The following change is recommended before acceptance of the manuscript.
- In the introduction, when the authors described the p53 family members with their functional domains, it would be useful to accompany the text with a figure. For the researchers of the p53 family field it could be redundant, but an occasional reader of the topic would find this initial figure useful to follow the rest of the paragraphs.
Author Response
Dear Reviewer.
Thank you very much for reading our manuscript and for you positive reply..
We incorporated a new Figure 1.
Sincerely.
Julian Rozenberg, PhD.